# Revived Amplicon Sequence Variants Monitoring in Closed Systems Identifies More Dormant Microorganisms

**DOI:** 10.3390/microorganisms11030757

**Published:** 2023-03-15

**Authors:** Ya-Xian Lu, Wei Deng, Fu-Liang Qi, Xiao-Yan Yang, Wen Xiao

**Affiliations:** 1Institute of Eastern-Himalaya Biodiversity Research, Dali University, Dali 671003, China; 2Collaborative Innovation Center for Biodiversity and Conservation in the Three Parallel Rivers Region of China, Dali 671003, China; 3The Provincial Innovation Team of Biodiversity Conservation, Utility of the Three Parallel Rivers Region from Dali University, Dali 671003, China; 4Yunling Black-and-White Snub-Nosed Monkey Observation and Research Station of Yunnan Province, Dali 671003, China

**Keywords:** gene function prediction, new method, high-throughput sequencing, microbial distribution, community structure, dormant microorganisms

## Abstract

The large number of dormant microorganisms present in the environment is an important component of microbial diversity, and neglecting dormant microorganisms would be disruptive to all research under the science of microbial diversity. However, current methods can only predict the dormancy potential of microorganisms in a sample and are not yet able to monitor dormant microorganisms directly and efficiently. Based on this, this study proposes a new method for the identification of dormant microorganisms based on high-throughput sequencing technology: Revived Amplicon sequence variants (ASV) Monitoring (RAM). Pao cai (Chinese fermented vegetables) soup was used to construct a closed experimental system, and sequenced samples were collected at 26 timepoints over a 60-day period. RAM was used to identify dormant microorganisms in the samples. The results were then compared with the results of the currently used gene function prediction (GFP), and it was found that RAM was able to identify more dormant microorganisms. In 60 days, GFP monitored 5045 ASVs and 270 genera, while RAM monitored 27,415 ASVs and 616 genera, and the RAM results were fully inclusive of the GFP results. Meanwhile, the consistency of GFP and RAM was also found in the results. The dormant microorganisms monitored by both showed a four-stage distribution pattern over a 60-day period, with significant differences in the community structure between the stages. Therefore, RAM monitoring of dormant microorganisms is effective and feasible. It is worth noting that the results of GFP and RAM can complement and refer to each other. In the future, the results obtained from RAM can be used as a database to extend and improve the monitoring of dormant microorganisms by GFP, and the two can be combined with each other to build a dormant microorganism detection system.

## 1. Introduction

Microorganisms are widely distributed in the Earth’s ecosystems, supporting a variety of ecosystem services, including nutrient cycling and climate regulation [1,2]. They are essential for maintaining ecological stability [3,4]. It is crucial to comprehend and clarify microbial diversity and its spatial and temporal distribution in biology and ecology research [5,6].

The incomplete monitoring of dormant microorganisms can lead to the incomplete monitoring of microbial diversity. The monitoring of microbial diversity is fundamental to the study of microbial diversity and ecology. The incomplete monitoring of dormant microorganisms would therefore interfere with all research under the discipline of microbiology and microbial ecology. It has been shown that dormancy can influence the spatial and temporal dispersal of microorganisms and environmental selection outcomes and that ignoring dormant microorganisms can obscure researchers’ understanding of spatial and temporal distribution patterns of microorganisms, such as distance decay relationships [7,8,9]. In contrast, few previous studies have taken into account the effects of dormancy on microbial diversity and spatial and temporal distribution, which may be an important reason why researchers have struggled to make accurate observations and judgments about factors that contribute to the geographic distribution patterns of microorganisms. In addition, dormancy is an important process for maintaining biodiversity, and dormancy reduces the extinction rate of microorganisms, thus maintaining the microbial diversity of the island [7]. Neglecting dormant microorganisms will therefore prevent accurate studies of microbial diversity and make it difficult to understand the spatial and temporal distribution patterns of microbial diversity and the mechanisms that maintain them.

The monitoring of dormant microorganisms is a prerequisite for addressing the above issues in order to aid the successful conduction of microbial diversity studies. The development of high-throughput sequencing technologies has greatly advanced microbiology; yet, the identification of dormant microorganisms is still lagging behind. The three methods listed below are frequently used to keep track of dormant microorganisms in samples: 1. To determine the physiological status of microorganisms in a sample, rRNA:rDNA ratios are utilized. By calculating the rRNA:rDNA ratio, one may determine the proportion of active and inactive microbial groups in a sample because the rRNA concentration is inversely correlated with the metabolic status of the microorganisms, and rDNA can serve as a reference for the total microorganism population. Sebastian Loeppmann et al. used the ratio of rRNA to rDNA in various soil samples to determine the physiological condition of microorganisms across multiple soil types [10], In addition, some scholars have studied the diversity and community structure changes of bacterial communities through rRNA and rDNA analysis [11,12]. 2. Calculating the rates of microbial respiration. By estimating the total oxygen consumed or carbon dioxide emitted by the microbes, one can determine the respiration rate of the organisms and characterize their biomass and microbial activity [13]. 3. Identifying dormant genes of microorganisms in samples using gene function prediction (GFP). GFP was utilized by Qicheng Xu et al. to find dormant potential microorganisms in samples that had genes related to spore production [14].

In the first method mentioned above, determining the rRNA-to-rDNA ratio can indicate sample activity and the total microbial biomass in the sample while also revealing information about the different physiological reactions of the microorganisms and soil characteristics. However, some scientists have discovered that rRNA does not accurately reflect the level of microbial action. For instance, when evaluating earlier research that used rRNA to characterize active microorganisms, Steven J. Blazewicz and colleagues concluded that rRNA is not a valid marker of active microorganisms [15]. Thus, the ratio of rRNA to rDNA does not accurately reflect the presence of dormant microorganisms. In the second method, soil microbial respiration reflects the total number and activity of microorganisms in the sample and can be an important indicator of soil quality and the response of soil microorganisms to environmental change. However, when dormant microorganisms are to be identified in a sample, the measurement of soil microbial respiration does not provide accurate information on dormant microorganisms. The third method, in contrast, uses GFP to identify dormant microorganisms more directly and successfully. By detecting dormant genes (such as spore-producing genes) and evaluating the dormancy potential of the microorganisms in the sample, GFP can determine the presence of dormant microorganisms in a sample [16,17]. However, in cases where microorganisms are dormant and in low abundance, GFP is also limited by high-throughput sequencing techniques for detecting their presence. Therefore, it is necessary to investigate a new method that can reliably and effectively find dormant microorganisms.

Due to changes in the environment, resources and other factors, dormant microorganisms can appear in the environment in the form of revived–dormant–revived. Therefore, based on the characteristics of dormant microorganisms, this study proposes a new method for identifying dormant microorganisms: Revived Amplicon sequence variants (ASVs) Monitoring (RAM). The ASVs in the presence–disappearance–presence and absence–presence states are dormant microorganisms when monitoring is repeated under this method on a sample kept in confinement (excluding exogenous microorganisms) (Figure 1). To verify the feasibility of the above method, this study set up a homogeneous and airtight microcosm using pao cai soup and carried out repeated monitoring of the bacterial community in the pao cai soup for 60 days, describing the dormancy dynamics of the bacterial community at 26 timepoints and comparing the results obtained with functional gene predictions. We conjecture that RAM will identify more dormant microorganisms than GFP over a 60-day monitoring cycle and that the two will form a consistent observation of changes in dormant microbial stages. Similarly, when soil or water samples are collected and stored in confinement, the presence of dormant microorganisms in the samples can be monitored with RAM. We expect that the RAM proposed in this study will outperform GFP.

## 2. Materials and Methods

### 2.1. Preparation of Pao Cai Soup

First, 35 kg of white radish (*Raphanus sativus*), 35 kg of cabbage (*Brassica oleracea*), 2 kg of chili pepper (*Capsicum frutescens*), 1 kg of ginger (*Zingiber officinale*), 1 kg of peppercorns (*Zanthoxylum bungeanum*), 2.5 kg of rock sugar, and 210 kg of cold boiled water (containing 6% salt) were packed into a ceramic jar. After 7 days of natural fermentation at room temperature, the pao cai was filtered out with sterile gauze to obtain 200 kg of pao cai soup [18]. To ensure an even distribution of microorganisms in the soup, the soup was mixed well and then left to rest for 12 h; the supernatant was taken, and the soup was left to rest for 12 h again. The pao cai soup was divided into sterile glass bottles and set three replicates. After the air was removed, the glass bottle was sealed with a sterile sealing film and cultured in an incubator at 25 °C.

### 2.2. Sampling and High-Throughput Sequencing

After dividing the pao cai soup, an original sample was collected for sequencing. The samples were sampled daily for days 1–10 of incubation, every 2 days for days 10–30 of incubation, and every 5 days for days 30–60 of incubation. The samples were collected 50 mL at a time, centrifuged at 8000 rpm for 10 min, and stored in a refrigerator at −80 °C. After all the sampling was completed, the samples were sent to Shenzhen Microcomputer Technology Group Ltd. (Shenzhen, China) for sequencing analysis. Total DNA was extracted from the samples according to the E.Z.N.A.^®^ soil kit (Omega Bio-Tek, Norcross, GA, USA) instructions. DNA concentration and purity were measured using NanoDrop2000, and DNA extraction quality was measured using 1% agarose gel electrophoresis. Sequence amplification and sequencing were performed using the extracted total DNA as a template and the 338 F (5′-ACTCCTACGGGAGGCAGCAG-3′) and 806 R (5′GGACTACHVGGGTWTCTAAT-3′) primers for PCR amplification of the bacterial V3–V4 variable region [19]. The PCR conditions were: 3 min of pre-denaturation at 95 °C, 27 cycles (30 s of denaturation at 95 °C, 30 s of annealing at 55 °C, 30 s of extension at 72 °C), and 10 min of extension at 72 °C (PCR Instrument: ABI GeneAmp^®^ Model 9700). To ensure the complete amplification of the target region. The amplification system was 20 μL, 4 μL 5*FastPfu buffer, 2 μL 2.5 mM dNTPs, 0.8 μL primers (5 μM), 0.4 μL FastPfu polymerase, and 10 ng DNA template. PCR products were extracted from a 2% agarose gel using the AxyPrepDNA Gel Extraction Kit. PCR amplification and sequencing were performed by Shenzhen Ltd., and the sequencing platform was the NovaSeq 6000 PE250 platform. The PCR products were recovered using a 2% agarose gel, purified using the AxyPrep DNA Gel Extraction Kit (Axygen Biosciences, Union City, CA, USA), eluted with Tris-HCl, and detected by 2% agarose electrophoresis. Libraries were constructed using the Illumina TruSeqDNA PCR-Free Library Preparation Kit (Axygen Biosciences, Union City, CA, USA) library construction kit. The libraries were then sequenced using the NovaSeq 6000 PE250 platform [20]. The final ASV tables were obtained after the quality control, trimming, denoising, splicing, and removal of chimaeras using the Qiime2 DADA2 plug-in. The representative sequences of the ASVs were then aligned to the Greengenes database to obtain a taxonomic table of the species. All non-bacterial sequences were removed using the Qiime2 feature-table plug-in and analyzed using Qiime2 Core-Diversity, R Studio 4.1.2 for alpha diversity and variance testing [21].

### 2.3. Data Processing

Based on the obtained gene sequences, the data were processed and analyzed under two methods: GFP and RAM. Data were screened and processed using Excel; relative gene abundance was calculated using R 4.2.2, and non-metric multidimensional scale analysis was used to assess the influence of different dormant microbial identification methods on the bacterial community structure [22]. The vegan package in R 4.2.2 was used to make columnar stacking charts.

## 3. Results

### 3.1. Comparison of the Number and Genus of ASVs Identified by the Two Methods

The number of ASVs with dormant genes reached 54.9% on day 9. During 10–20 days, the number of ASVs decreased from 52.0% to 13.1%, while, during days 21–60, the number of ASVs increased again, reaching 34.4%. The top 20 taxonomic genera were chosen based on relative abundances from a total of 270 bacterial taxonomic taxa, including Exiguobacterium, Enterococcus, Rahnella, and Brochothrix (Figure 2b). The fluctuation ASVs monitored by the RAM were highly variable during days 1–10 and reached a maximum on day 9, with a gradual decrease in the number of fluctuation ASVs during days 12–24 and no further significant changes during days 26–60 (Figure 2c). At the genus level, a total of 616 bacterial taxonomic genera were monitored in the RAM, selecting the top 20 taxonomic genera in relative abundances, such as Enterobacter, Exiguobacterium, Citrobacter, Gluconacetobacter, and Rahnella (Figure 2d). In addition, both methods monitored dormant species with unannotated ASVs that were classified as candidatus. A total of 270 genera monitored by GFP had between 0.0% and 0.4% of candidatus, and 616 genera monitored by RAM had between 25.8% and 45.3% of candidatus.

A total of 346 genera appeared only in RAM’s results for identifying dormant microorganisms, including Citrobacter, Gluconacetobacter, Aerococcus, and Thiobacillus.

### 3.2. Consistency of the Two Methods

The ASVs identified by GFP and RAM wed a four-stage distribution pattern under a 60-day monitoring cycle. The distribution is shown in Figure 3, with the ASVs identified by GFP divided into four phases at days 1–6, 7–10, 12–30, and 35–60. The ASVs identified by RAM were divided into four phases at days 1–6, 7–10, 12–30, and 35–60 (Figure 3).

The differences in the structure of the pao cai bacterial communities under various temporal stages were analyzed using non-metric multidimensional scaling analysis (NMDS) based on the Bray–Curtis distance (Figure 4), with a model stress function value of Stress < 0.2, indicating a reliable method. The results showed substantial differences between the bacterial communities, and the distribution of samples under the four stages was largely independent.

The number of bacterial taxa monitored by the GFP method was significantly lower than that of those monitored by the RAM, and the bacterial taxa monitored by the RAM covered all the bacterial taxa monitored by the GFP method (Figure 5).

The number of bacteria genera monitored by GFP and RAM showed a trend of substantial growth from the 1st day to the 10th day and gradually slowed down from the 11th day to the 60th day (Figure 6a). With the increase in sampling times, the number of bacterial genera also increased gradually. Similarly, the number of bacterial genera increased rapidly during the 1st to 10th sampling times (Figure 6b).

RAM identified 32.4% of dormant microorganisms on day 5; GFP identified 32.1% of genera on day 9. RAM identified 32.7% of genera at sampling time combinations of days 1 and 5 and 51.9% of genera at days 1, 5, and 9. GFP identified 32.5% of genera on days 1 and 9 and 50.9% of genera on days 1, 5, and 9 (Figure 7).

## 4. Discussion

The closed system in this study was free from the dispersal of foreign microorganisms and would not have affected the results. Any new microbes that emerged should be dormant in the system, while all microorganisms identified were from the original samples [23]. During the 60 days of monitoring, GFP detected a total of 5045 AVSs and 270 bacterial taxa; RAM detected a total of 27,415 ASVs and 616 bacterial taxa.

The ASV monitored by GFP and RAM showed a four-stage distribution pattern over the 60 days of long-term monitoring, and both differed significantly. During the 10-day short-term monitoring, RAM monitored 431 genera, or 69.9% of the total dormant genera (616), and GFP monitored 189 genera, or 70.0% of the total dormant genera (270), while the number of dormant microorganisms monitored by both methods showed a small increase thereafter. The above results show that GFP and RAM are consistent and confirm the feasibility of the RAM method proposed in this study. In contrast, RAM monitored a greater abundance of dormant microorganisms than GFP. In terms of the number of ASVs, RAM monitored 22,370 more ASVs than GFP; in terms of the number of bacterial genera, RAM monitored 346 more genera than GFP, and the results of RAM monitoring fully encompassed those of GFP. Therefore, we believe that the use of RAM to identify dormant microorganisms is feasible and effective.

Although GFP has the advantage of being able to identify dormant microorganisms in a single sequencing [24], GFP can only monitor 3.70% of dormant microorganisms in a single sequencing. RAM was able to monitor 4.5% of dormant microorganisms on days 1 and 2, 23.5% on days 1 and 3, 23.2% on days 1 and 4, and 32.7% on days 1 and 5. In contrast, 319 genera were monitored on days 1, 5, and 9, representing 51.7% of the total. These results show that, in the short term, RAM can monitor more than 20% of dormant microbes with just one additional sequencing and identify more than half of the dormant microbes with two additional sequencings. Therefore, when controlling the cost of sequencing, better results can also be achieved by using RAM to monitor dormant microorganisms. For the pao cai samples used in this experiment, we suggest that sampling can be selected on days 1 and 5 or days 1, 5, and 9.

The dormant microorganisms identified by RAM are those that have actually gone dormant, making it more useful to carry out studies on the ecological functions of dormant microorganisms, which is also an advantage of RAM. The microorganisms detected by GFP are those with spore-producing genes that are active when they are detected, while those that are dormant cannot be detected because their abundance is too low [25]. In addition, microbial dormancy is not always in the form of spore production; it is also common for microorganisms to regulate their metabolic levels and change their morphology and size [26,27]. The identification of dormant microorganisms by spore-producing genes alone would therefore lead to an inaccurate understanding of dormant microorganisms, which would also hinder research into the ecological role of microbial dormancy.

It is important to note that the RAM and GFP methods are not alternatives or opposites to each other but rather complementary to each other. RAM can detect more dormant microorganisms than GFP, and many microorganisms that cannot yet be cultured and identified (candidatus) can also be identified by RAM. Therefore, the dormant microorganisms identified by RAM can enrich the dormant microorganism species library, and future research at the dormant gene level can be carried out on the dormant microorganisms identified by RAM, which will enrich the gene database of GFP and thus facilitate the recognition rate of dormant microorganisms by GFP. We propose combining both the RAM and GFP methods for monitoring dormant microorganisms in the future, first using more RAM to enrich the gene database for some time to come and then using GFP to reduce costs, thus enabling the efficient monitoring of dormant microorganisms.

Finally, both the RAM and GFP results obtained indicated that dormant microbial communities in pao cai storage appeared to change in phase. These results may herald a role for dormancy in the maintenance of microbial diversity. The dormancy recovery of microorganisms is a process that occurs continuously with community succession. Microorganisms with the ability to become dormant in a sample will choose to become dormant to ensure their presence in the system as conditions become unsuitable, while those that do not have the potential to become dormant gradually become extinct, leading to changes in community structure. Microorganisms that are dormant at this time act as seed banks to replenish the active microbial community in order to maintain the stability of the community structure and the function in the system [28]. This study provides a methodological basis for the study of the above ecological processes, and the use of the RAM proposed in this study may have the opportunity to further elucidate the ecological functions of dormant microorganisms.

## 5. Conclusions

Dormancy has a significant role in microbial community building and spatial and temporal distribution, and future research on dormant microorganisms should be intensified in view of its ecological function and its impact on microbial diversity assessment. In this study, we propose a new method for monitoring dormant microorganisms (RAM), which has been shown to be feasible and effective when compared to the currently used GFP. We suggest that future research should first use RAM to enrich the dormant microbial database and then combine RAM with GFP to reduce costs and improve efficiency.

## Figures and Tables

**Figure 1 microorganisms-11-00757-f001:**
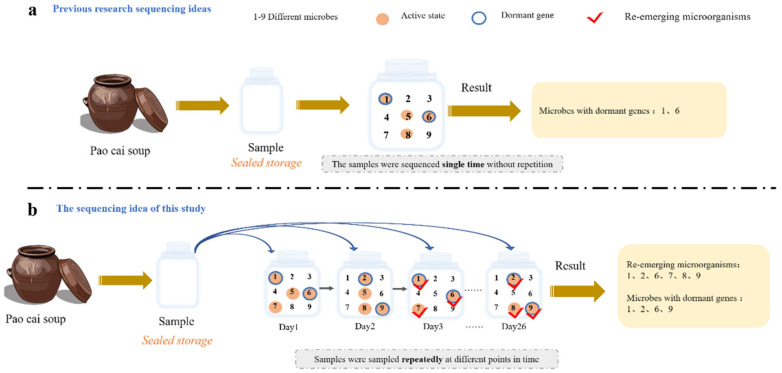
Dormant microorganism identification methods. (**a**) GFP: contains functional genes associated with dormancy in microbial DNA in samples that can be detected by high-throughput sequencing, but only when the microorganism is active and contains dormant genes. (**b**) RAM: multiple sampling and high-throughput sequencing of the original sample, allowing fluctuations in the microorganisms in the sample to be observed over time.

**Figure 2 microorganisms-11-00757-f002:**
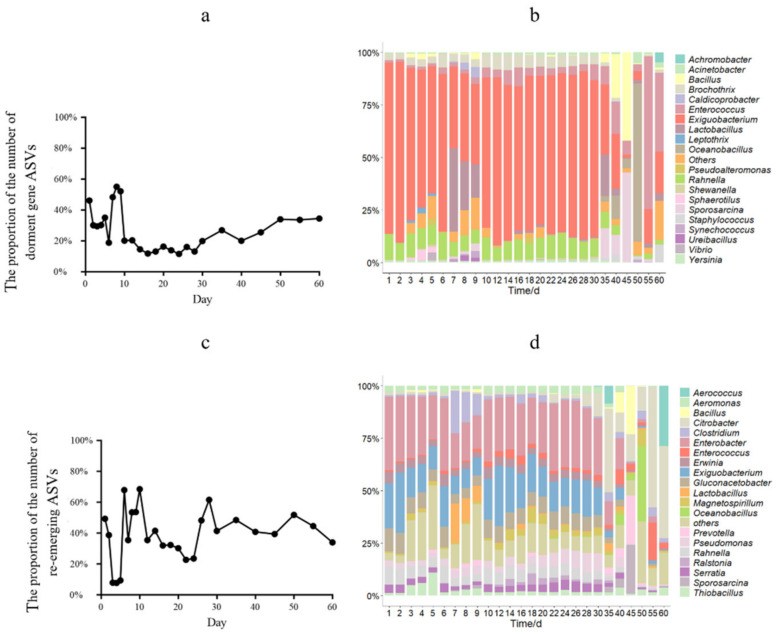
Number and genus of ASVs identified in 60 days. (**a**) Change in the proportion of ASVs identified by GFP; (**b**) genus identified by GFP; (**c**) change in the proportion of ASVs identified by RAM; (**d**) genus identified by RAM.

**Figure 3 microorganisms-11-00757-f003:**
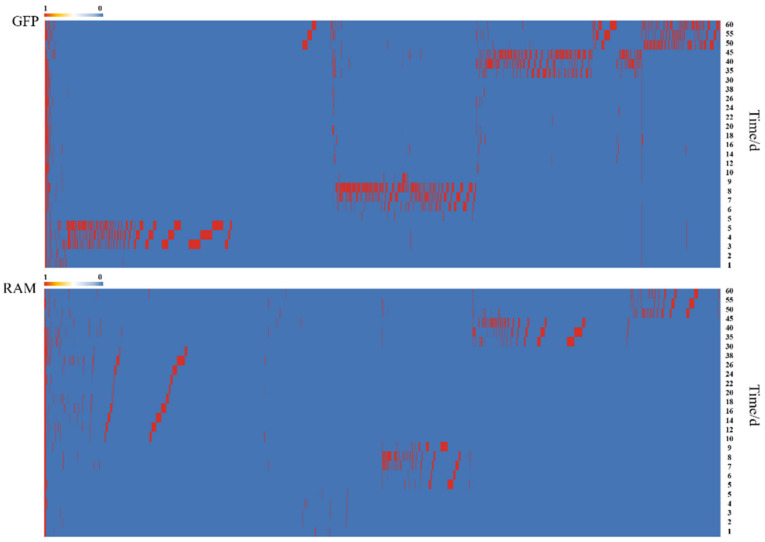
Distribution of ASVs identified in 60 days.

**Figure 4 microorganisms-11-00757-f004:**
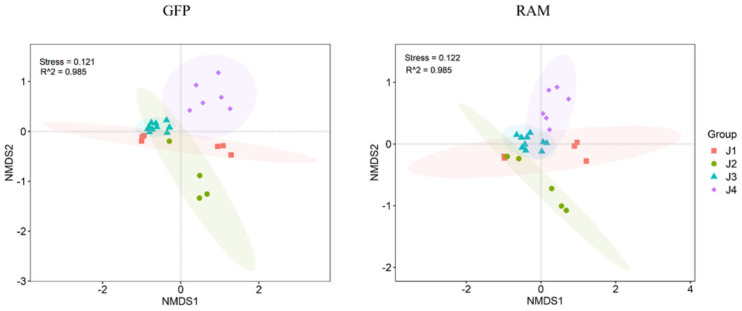
NMDS analysis of differences in bacterial community composition at four stages under two methods.

**Figure 5 microorganisms-11-00757-f005:**
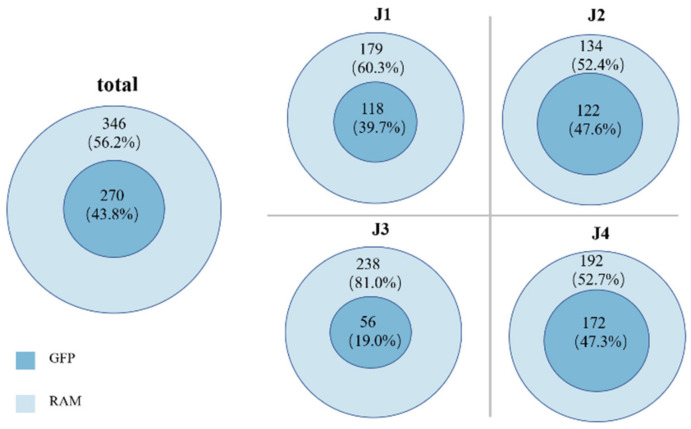
The difference in the amount of GFP and RAM in the four stages.

**Figure 6 microorganisms-11-00757-f006:**
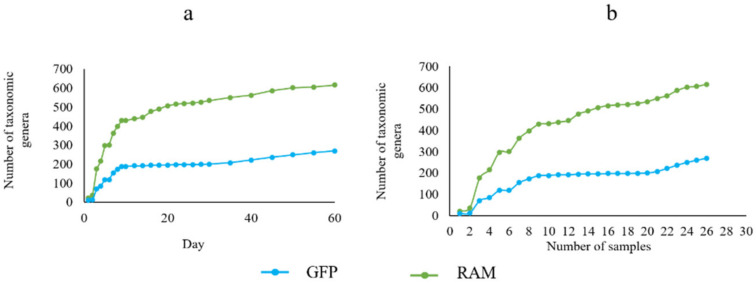
(**a**) The trend of the number of dormant bacteria species with the monitoring time; (**b**) The trend of the number of dormant bacteria species with sampling times.

**Figure 7 microorganisms-11-00757-f007:**
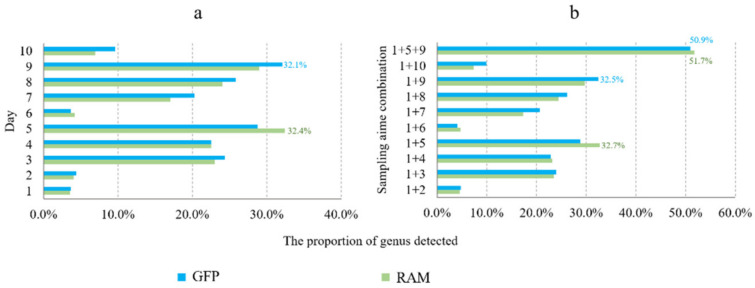
Genera of dormant microorganisms detected at different combinations of sampling times. (**a**) Proportion of bacterial genera detected by GFP and RAM to total genera in days 1–10; (**b**) Proportion of bacterial genera to total genera detected at different combinations of days within 10 days.

## Data Availability

The raw sequence data reported in this paper have been deposited in the Genome Sequence Archive (Genomics, Proteomics & Bioinformatics 2021) in the National Genomics Data Center (Nucleic Acids Res 2022), the China National Center for Bioinformation/Beijing Institute of Genomics, and the Chinese Academy of Sciences (GSA: CRA008411), and they are publicly accessible at https://bigd.big.ac.cn/gsa/browse/CRA008411, accessed on 1 March 2023.

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
