# Peer review of "Revived Amplicon Sequence Variants Monitoring in Closed Systems Identifies More Dormant Microorganisms"

_microorganisms, 2023, doi:10.3390/microorganisms11030757_

Round 1

Reviewer 1 Report (Previous Reviewer 1)

The authors have made changes that I recommended in the previous version. However, some changes needed are:

• Acronyms should not be described more than once (you do it 3 times).

• I recommend that the document should be written in impersonal.

• The conclusion section needs to be improved.

• In section 2.3 (Data Processing) I don't think the internet links are necessary. In my view, they are not important.

• Line 174 what does Vegan mean? 

Author Response

Dear reviewer,

We appreciate your thoughtful analysis and helpful criticism of our paper. We have revised the manuscript based on your suggestions. Here are our answers to the questions you raised.

Comment 1: Acronyms should not be described more than once (you do it 3 times).

Response: We checked the parts of the text where acronyms appeared and removed the abbreviation RAM from the title, retaining the abbreviations ASV, RAM, and GFP in the abstract and body of the text. The abstract and text are two separate sections and we felt it was necessary to retain the abbreviations.

Comment 2: I recommend that the document should be written in impersonal.

Response: Thank you for your suggested changes, we have checked the full text and made changes.

Comment 3: The conclusion section needs to be improved.

Response: We have rewritten the conclusion section in the hope that it meets your requirements.

Comment 4: In section 2.3 (Data Processing) I don't think the internet links are necessary. In my view, they are not important.

Response: Thanks to your suggested changes, we have removed the Internet links to LN172 and LN175.

Comment 4: Line 174 what does Vegan mean?

Response: Vegan is a package in R. We have added "package" after LN173 "Vegan" in the revised version.

All the best,

Wen Xiao

Institute of Eastern-Himalaya Biodiversity Research, Dali University

Reviewer 2 Report (Previous Reviewer 2)

Microorganisms-2201305

Revived Amplicon sequence variants Monitoring (RAM) in closed systems identifies more dormant microorganisms

Lu et al. have revised their manuscript and satisfied my previous concerns. I believe this manuscript is ready for publication, after some minor proofreading.

Author Response

Dear reviewer,

Thank you for your careful review and constructive comments on our article. We have revised the manuscript based on your suggestions. Here are our answers to the questions you raised.

Comment 1: Revived Amplicon sequence variants Monitoring (RAM) in closed systems identifies more dormant microorganisms

Lu et al. have revised their manuscript and satisfied my previous concerns. I believe this manuscript is ready for publication, after some minor proofreading.

Response: Thank you very much for your approval, we have made some proofreading to the full text.

All the best,

Wen Xiao

Institute of Eastern-Himalaya Biodiversity Research, Dali University

Round 2

Reviewer 1 Report (Previous Reviewer 1)

The authors improved the document with my suggestions.

I recommend the publication.

This manuscript is a resubmission of an earlier submission. The following is a list of the peer review reports and author responses from that submission.

Round 1

Reviewer 1 Report

 Some changes needed are:

 - It does not have the conclusions section.

-I recommend increasing the number of references, mainly on the methodology and the discussion of results. The references used only present one citation in methodology and one citation in the discussion (very few).

-I recommend expanding the discussion and increasing the number of references.

-The acronyms must be specified from the Abstract, (ASV).

Reviewer 2 Report

Microorganisms-2201305

Revived Amplicon sequence variants Monitoring (RAM) in closed systems identifies more dormant microorganisms

Lu et al. propose and test a novel method for discovering dormant microorganisms in microbial communities. This is an important question in microbial ecology, with most current methods poorly suited to distinguishing between active and inactive but alive microbes, and a great many microbial taxa have been shown to be capable of forming a resistant resting stage during conditions not suited to their active growth.

The proposed method differs from others primarily in comparing community sequencing results across multiple time points, a difference in both sampling intensity and data analysis. Organisms detected in later samples but not earlier samples are inferred to have revived, indicating they were dormant during the earlier time points. This inference relies on certain assumptions being met, such as no possibility of new taxa immigrating into the study system. This manuscript studied a sealed, fermented soup under laboratory conditions and thus meets this assumption. Other systems, such as soil or water, would need to be isolated and incubated in the laboratory to be studied by RAM, which introduces additional variation compared to a natural soil or water system with continuous interaction with other microbial communities.

The method does seem like a useful advance in this area. Unfortunately, some aspects of the data analysis seem to render the results meaningless. The inclusion of the term “candidatus” as a genus name in the evaluation of diversity captured by the two methods compared here means the entire results section should be re-analysed. Any similarities or differences between the two methods are confounded or concealed by this error. Fortunately, this re-analysis should be straightforward, and the various quantities reported can be updated.

It might be useful to include a flow chart or brief written description of the procedure employed, from isolation of a sample through to comparisons across sampling times, to clarify how RAM differs from other approaches.

This manuscript is largely missing a Discussion section. A few brief notes suggest some possible ways to improve and develop this method and integrate it into current research in microbial ecology, yet there is little detail and most of the Discussion is simply a re-statement of the main quantitative results.

In short, the authors need to re-analyse the data with the mistaken term candidatus excluded, and provide more insight into how this novel method may be best employed by other researchers.

Abstract

Some spelling and grammatical issues here, such as “The ASVs of revive was compared” (LN20) – please define ASV, and I suggest the words change to “The ASVs revived were compared”

Also, the plural of genus is genera, as in “GFP found 271 genera”

The keywords might usefully also include “Dormant microorganisms” or similar.

1. Introduction

LN41-42: “Current research is still disputed” yet the citation, [7], is a paper from 2006. Microbial ecology is a fast-enough-developing field that surely there exists a more recent relevant publication.

LN72-74: The citations here are a bit awkward. Starting the sentence with a specific lead author is fine, but the collection of citations at the end of the sentence seems to imply all three publications have the same first author, which is not the case for [16, 17, 18]. I suggest inserting the specific citation to go with that author immediately after the “et al.” in the sentence, and adding the other two citations at the end.

In the paragraph from LN100 to LN112, in which the novel method is described, there is some confusion about how this process works through time. Continuous sampling implies sampling is undertaken by a continuous flow of liquid media through a detector, as in the case of some chemical analyzers that continuously monitor pH or the concentration of some solute. The proposed method, RAM, seems to be rather a series of discrete sampling events, highly replicated but not continuous.

2. Materials and Methods

LN156: this is the first time that “ASV” is defined. It should be defined in the Introduction.

LN165: please provide citation for R.

3. Results

LN170: how much is “a lot” of fluctuation? Compared to what?

LN180: “Candidatuss” is misspelled (only one s at the end) and, more importantly, is not a genus of microorganism. The term Candidatus is used to describe organisms that have not yet been formally assigned a scientific name, though most often a proposed or putative name is included after the term candidatus. A database search for genera in common between two samples that employs the term “candidatus” would return a large number of false positives.

The large fraction of microbes assigned to psuedo-genus Candidatus illustrated in Figure 2 D is an example of this kind of serious problem. There is no indication that any two “Candidatus” species are in fact in any way taxonomically or phylogenetically related.

The large difference in proportion of identifications as “Candidatus” between the two methods, where these pseudo-genera are much less common in the GFP analysis, is itself interesting in this context – is there a difference in how genes are annotated (especially to the category of dormancy genes) in organisms with a recognised scientific name (anything other than candidatus) and those that have only recently been identified, isolated, and characterised?

I think it is also very interesting that the seemingly-abundant psuedo-genus Candidatus is absent from the top 20 relatively abundant genera.

LN186: “using the wave sequence’s continuous monitoring strategy” – again, is this actually continuous, or is it instead highly repeated, in a large series of discrete sampling events? This seems to be the first mention of the term “wave sequence” – is this another name for the proposed novel method, otherwise known as RAM?

LN216-218: The trends in Figure 7 do not appear to reach “leveled off” by either method. The lines are still trending upwards by Day 60 or sampling number 36. There does appear to be a slowing of the rate of increase starting around Day 10 or 10th sampling (which is functionally the exact same thing – the tenth sample was collected on Day 10, wasn’t it?).

LN218: What statistical test resulted in the judgement that the trend had “stopped significantly increasing”? The word “significant” implies a statistical test and, most often, an associated p-value or similar means of making such a judgement.

LN222-226: The percentages here presumably refer to some maximum, 100%. Is the number of dormant microorganisms at the end of the experiment (Day 60) taken to be that 100%?

4. Discussion

Much of the Discussion appears to have been directly copied from the abstract (or vice-versa), even down to the errors such as the missing space on LN236 “271genus” that directly mirrors the same error on LN24.

I would hope for much more in a Discussion section. This manuscript proposes and describes a test of a novel approach to studying dormant microorganisms – surely there is more to discuss than simply a series of raw numbers and percentages. Those are Results – one method was better than the other – not Discussion.

There is a small amount of reflection and future work suggestions in the second paragraph, but these ideas could be considerably expanded. Under what circumstances might GFP be better suited than RAM, if any? How can the results of RAM be fed back into improving GFP – would the RAM-identified organisms be subjected to more intense scrutiny to discover their dormancy-related genes? Are there research areas where RAM seems to be particularly well-suited, beyond fermented foods?

Typos and other minor issues.

LN47: rather than varying the vocabulary, I suggest using the widely-employed term “community assembly” instead of “community construction”.

LN105: I think the word “repeatedly” is a better fit than “constantly” – constant sampling implies a continuous process, not something repeated 26 times.

LN128: “Divide pao cai soup into” is formatted as a command, as in instructions. The appropriate way to write a Methods section is to use past tense, as in “The pao cai soup was divided into sterile glass”. Descriptive present tense, as on LN129-130 is also acceptable.

LN135: delete the comma , after “After all”

LN143-145: this is not a complete sentence. Maybe start with “The PCR conditions were:”

LN199 (caption for Figure 4): “over 60 days” or “in 60 days” rather than “on 60 days”.